

# Interoceptive awareness and beliefs about health and the body as predictors of the intensity of emotions experienced at the beginning of the pandemic

Aleksandra Modzelewska and Kamil K. Imbir

Faculty of Psychology, University of Warsaw, Warsaw, Masovian, Poland

## ABSTRACT

**Background:** The COVID-19 pandemic is a type of stressful event which might have an impact on psychological state. A prolonged threat of getting a serious, contagious illness is expected to be associated with an increase of negative emotions and, conversely, with a decrease of positive emotions. As the stressor is strongly linked to health and the body, we decided to investigate what types of factors related to body perception and appraisal are associated with different types of reported emotions. The purpose of the study was to verify the associations between three types of variables: interoceptive awareness as described by Mehling and colleagues (2012a, 2018), negative beliefs about health and body, and different types of emotions.
**Methodology:** A Multidimensional Assessment of Interoceptive Awareness questionnaire was applied to evaluate interoceptive awareness. The declared emotional state was diagnosed with a list of 20 emotions–divided by valence and origin. Additionally, a list of 10 negative beliefs about health and body was used. The study was held in a correlational schema with 299 subjects recruited *via* the social media platform Facebook who took part in an online survey.
**Results:** The study revealed that the scales of Self-Regulation and Trusting are primarily associated with negative automatic and reflective emotions and positive automatic emotions. Furthermore, the Self-Regulation, Trusting dimensions of interoceptive awareness predict an intensity of emotions categorised on the basis of valence and origin. In addition, negative beliefs about health and the body provided an adequate explanation of the variance of most of the types of emotions experienced during the pandemic.
**Conclusions:** Factors related to body perception, such as interoceptive awareness and negative beliefs about health and body provide a significant contribution to explaining emotional state at the beginning of the pandemic.

Corresponding author
Aleksandra Modzelewska,
a.modzelewska4@uw.edu.pl

# INTRODUCTION

## The ecological background of the study–the context of the pandemic

This study was run during the beginning of the COVID-19 pandemic in Poland, from 27 March until 15 April 2020. The severity of the pandemic at that time can be described

by the following indicators: on 27 March 1,389 cases of COVID-19 had been diagnosed and 16 patients had died, whereas on 15 April, 7,582 cases were diagnosed, there had been 286 fatalities and 668 had recovered. During that period, the government implemented many restrictions limiting ordinary social activities. From 1 April, the government announced a ban on movement to reduce the rate of spread of the epidemic. The "lockdown" strategy was used for the first time in Poland. It can be said that the study was conducted during a period of significant restrictions on civil liberties and a serious growth of the pandemic; additionally, the novelty of the situation could have provoked increased stress. A pandemic is a stressful event, which significantly influences individuals' emotional states.

In the current study, we would like to focus on the subject's declared emotional state during the first stage of the pandemic, and to verify the factors that may modify this state. Interoceptive awareness is related to emotional experience (*e.g.*, *Barrett et al., 2004*; *Hanley, Mehling & Garland, 2017*), and therefore it should modify emotional reactions to threatening circumstances. At the same time, exposure to a serious, contagious disease involves a fear for one's health (*Person et al., 2004*). Accordingly, beliefs about health and the body were also included in the study design as a factor influencing emotional state during the first stage of the COVID-19 pandemic.

The pandemic situation can be highly stressful. *Folkman & Lazarus (1985)* defined stress as a type of relationship between humans and the environment when resources are exceeded. An individual experiencing an overloading situation might feel the threat of potential harm or loss, which is a stress component (*Folkman & Lazarus, 1985*). It is the perception of the situation in terms of a threat, not a challenge, that causes a negative emotional response (*Dienstbier, 1989*). Importantly, research demonstrated that threat perception is associated with cardiac and vascular reactivity (*Tomaka et al., 1993*). This is not the only manifestation of stress at the physiological level. The meta-analysis conducted by *Dickerson & Kemeny (2004)* revealed that acute psychological stressors are related to an increase in cortisol level; however, the reaction varied across conditions. In addition, stress also has a significant impact on mental health. Several meta-analyses confirmed the association between stress and depression (*e.g.*, *Kessler, 1997*; *Mazure, 1998*; *Monroe & Hadjiyannakis, 2002*; *Paykel, 2003*). Essentially, there is evidence confirming the linear connection between the severity and amount of stressful events and vulnerability to depression (*Kendler, Karkowski & Prescott, 1998*).

Data on the current pandemic confirm that people presently experience more stress than before. Research done in January and February 2020 in China confirmed that 54% of respondents perceived the psychological impact of disruption caused by the COVID-19 pandemic as severe or moderate, 29% declared experiencing moderate or serious anxiety symptoms and 17% indicated a depressive reaction (*Cullen, Gulati & Kelly, 2020*). If we look at data describing past epidemics, we can conclude that this type of phenomenon is inherently a large emotional burden. In a paper about the psychological consequences of the influenza pandemic, *Perrin (2009)* indicated that although people do indeed address fear and new, unknown health dangers, the severity of anxiety experienced and the type of coping strategies are determined by individual differences and

might evolve during the epidemic. There are many examples in the empirical literature illustrating the influence of epidemics on mental health and emotional state. For example, during an outbreak of Legionnaires Disease in Japan, 17.6% of respondents reported feeling stressed, 13.7% reported anxiety and insomnia and 3.3% reported severe depression. Moreover, 3–4 months after the breakout, some patients still demonstrated mental health problems (*Tsuruta et al., 2005*). Similarly, in 18% of the study population, severe acute respiratory syndrome (SARS) in Hong Kong was associated with anxiety, post-traumatic stress disorder and depression symptoms (*Wu, Chan & Ma, 2005*). Taking into account the above empirical data, it can be seen that a pandemic is associated with various emotions and might cause a distress response with serious consequences for mental health. So, the question is, what kind of emotions are presently people experiencing, during the COVID-19 pandemic? To answer this question, we first need to define appropriate dimensions to describe emotions.

## Types of emotions–valence and origin components

To discuss the nature of the dimensions of emotion, it is first necessary to briefly define the concept of emotion. Some authors define emotions as a response (physiological, psychological and behavioural) (*Frijda, 2007*) to the evaluation of stimuli (*Jarymowicz & Imbir, 2015*) which might address both biological aspects (*Damasio, 2010*) and the cognitive interpretation of the ongoing situation from the perspective of an individual's goals and standards (*Clore & Ortony, 2010*). Consecutively, *Moors & Fischer (2019)* consider emotions as a product of a goal-directed mechanism, referring to adaptability. Emotions express a tendency to action directed towards the achievement of a goal. The appraisal theories of emotions emphasise the relevance of the stimulus in reference to goals (*Moors et al., 2013*). The appraisal of stimulus is a type of verification of the congruence of stimulus with goals and controllability of this goal, thus it influences behaviour. The proposed view on emotions is in line with an alternative approach to dual-processes in the action and emotion domain proposed by *Moors, Boddez & De Houwer (2017)*. The goal-oriented processes are considered a default (instead of stimuli-driven); moreover, these processes are not limited to the regulation of emotion, but are expected to strongly affect the formulation of action tendencies. The goal-oriented approach emphasises the role of possible control of aversive stimuli–when an individual is facing a choice between a fight or flight reaction; the choice is dependent on the expected utility of both reactions (*Moors et al., 2019*).

Taking into account the multitude of definitions and the variety of approaches (sometimes partially or completely contradictory to each other), one can get the impression that the construct of emotions still arouses some controversy, and the outlining of an unambiguous definition accepted by most researchers seems to be instinctive, albeit difficult to achieve.

To get deeper into the concept of emotions it is useful to address the key dimensions of this phenomenon. One of the basic dimensions of emotions is valence, a factor defined

as the pleasantness or unpleasantness of experiences (*Barrett, 1998*; *Lazarus, 1991*). Valence results from appraisal the of stimuli as positive, negative or neutral, thus reflecting the basic biological tendency of approaching and avoiding. This elementary dimension enables the description of the emotional state in terms of positive and negative emotions, thus responding to the basis of the need to define feelings experienced in terms of comfort and discomfort. It is common to both the theoretical scientific approach and the common typology of emotions.

While the dimension of valence is fundamental in most research on emotion, there are still limited data about the origin dimension. This dimension refers to the broader heart-mind dichotomy rooted in classical philosophy. Based on the origin typology proposed by *Jarymowicz & Imbir (2015)*, emotions can be divided into those resulting from automatic ("heart") evaluative processes that do not need explicit cognition to appear and those following reflective ("mind") evaluative processes that are based on extended cognition (*Jarymowicz & Imbir, 2015*). Automatic emotions result from internal drives and needs; they also refer to external incentives that play a crucial role in social and physical survival. They are thus related to the concept of biological value (*Damasio, 2010*), an automatic criterion of evaluation that gives an organism the sense of what is good for survival (*i.e.*, pleasant) or bad (*i.e.*, unpleasant). Automatic emotions are regulated by affect (*Zajonc, 1980*) and thus have a diffusive, holistic and homogeneous character (*Jarymowicz & Imbir, 2015*). The automaticity and reflectiveness in this approach are related to different degrees of cognition; automatic emotions need a minimal level of cognition to occur, in contrast to reflective ones. Nevertheless, the automaticity is not understood synonymously as absolute spontaneity. Reflective emotions are driven by intellectual, "mind" principles; their evaluation has a verbalised nature (*Reykowski, 1989*; *Strack & Deutsch, 2004*, *2014*). They are relatively heterogeneous and precise. The criteria of evaluation develop over the entire lifespan and are based on reflections concerning what is good or bad (*Jarymowicz, 2012*; *Jarymowicz & Imbir, 2015*).

The proposed theoretical conceptualisation of automaticity and reflectiveness in terms of emotion deliver an interesting framework; however, its postulates might draw some contraventions. For example, *Moors & De Houwer (2006)* question defining a concept of automaticity through categories such as unintentionality, uncontrollability, autonomy, stimulus dependency, goal independence, efficiency and high speed. Researchers indicate the overlapping character of these components and suggest investigating each feature separately. Moreover, based on theoretical analysis, authors conclude that the features mentioned have rather a dimensional nature, therefore cognitive and emotional processes should be located somewhere on a continuum.

Nevertheless, in the current study we used *Jarymowicz & Imbir's (2015)* view on emotions, treating automaticity and reflectiveness as opposite ends of a scale representing the engagement of explicit deliberation towards stimulation, leading to interpretation of the situation. In our opinion, this approach might offer a promising theoretical framework to investigate the possible diversity of emotions reported during the pandemic.

## Factors related to psychological attitude towards the body affecting emotional state

In the context of the pandemic, stress, which is naturally experienced within the body, seems to be strongly associated with concern about the condition of one's body, in other words, about one's health. Therefore, we might assume that an individual's perception and awareness of their body might influence their currently self-reported emotional state. One such factor might be cognitive beliefs about the body resulting from perception and evaluation of the body. A Health Beliefs Model delivers a kind of theoretical support for the importance of individuals' beliefs, health and behaviour (*Hochbaum, 1985*; *Rosenstock, 1966*, *1960*; *Rosenstock, Strecher & Becker, 1988*). This model was initially designed to explain the failures of prevention programmes relating to health, and then it was expanded to explain reactions to symptoms and diagnosed diseases. The model refers to value-expectancy theory and includes the perceived severity of illness, the perceived benefits, or action taken to reduce the threat of illness and promote health and perceived barriers referring to cost-benefit analysis if the actions taken involved any sacrifices (*Rosenstock, Strecher & Becker, 1994*). The role of beliefs in the approach to disease conceptualised in the model was empirically verified (see, for example, the meta-analysis conducted by *Harrison, Mullen & Green (1992)*). Furthermore, research proved that optimistic self-beliefs enhance health-protecting behaviour through setting goals, taking actions and building motivations (*Schwarzer, 1994*). Finally, the relationship between emotional state and beliefs can be addressed with the ABC model constituting one of the foundations in CBT and RET therapies. Based on the model, cognition mediates emotional reaction (*Ellis, 1991*).

By describing how people experience their bodies, a distinction can be made between the perceived sensations and the declared state. When a person suffers from the disease, the observed physiological dysfunctions are associated with symptoms reported by this person (*Pezzulo et al., 2019*). However, some fluctuations in this relationship can be noticed. To explain why the relationship between objective disease parameters and symptoms reported by patients may vary significantly, Pezzulo and colleagues (2019) proposed a new predictive processing coding perspective. In this conceptualization, the experience of symptoms is the product of not only interoceptive perception but also predictions of the brain about detected interoceptive signals. We can therefore conclude that the perception process is not independent of the previously encoded meanings. Apart from perception (defined as predictive coding), the model also includes the level of action understood as an active interference. The role of pre-existing (prior) information in the construction of newly occurring representation expresses some similarity to the influence of cognitive beliefs on the formulation of emotional response.

To discuss further the role of related factors in shaping emotional state, we going to describe the concept of interoceptive awareness. The interoception is currently one of the major topics in mental health and body-mind based interventions (*Mehling et al., 2018*). Recent approaches put a strong emphasis on the beneficial influence of the enhancement of body awareness, especially in individuals who are professionally involved

in mind-body therapies like yoga, tai chi or meditation (*Mehling et al., 2009*, *2012*, *2011*). Moreover, some indicate that an increase of interoceptive awareness observed in mindful awareness in body-oriented therapy (MABT) has a beneficial influence on an individuals' emotional state. It supports the ability to adequately detect and classify physiological sensations, therefore it enables an adequate reaction to stressful stimuli and appropriate regulation (*Price & Hooven, 2018*). Considering the literature cited, we drew a conclusion that interoceptive awareness should play a crucial role in explaining the emotions experienced. For a broader discussion of the topic of interoceptive awareness, some definitions will first be presented, and then the relationship between interoceptive awareness and emotions will be described.

Interoception can be defined as a process reflected in nervous system activity by which signals both from and within the body are detected, interpreted and integrated (*Mehling et al., 2018*), and which is also understood as a body-to-brain axis related to the state of the inner body and visceral organs (*Cameron, 2001*). *Quadt, Critchley & Garfinkel, (2018)* emphasise a basic function of interoception, which is keeping the body alive through the afferent channel of the brain and body interaction oriented to maintenance of homeostasis. *Tsakiris & Critchley (2016)* described interoception as a process of detecting and sensing internal states through the processing of inner body sensation. Considering the definition thus noted, it is worth emphasising, however, that awareness of the state of the inner body and visceral organs (*Cameron, 2001*) differs from detection, followed by interpretation and integration (*Mehling et al., 2018*). The understanding proposed by Mehling is a wider concept and seems to refer both to awareness of bodily psychological sensations and to the appraising interpretation occurring with such awareness (*Mehling, 2016*; *Mehling et al., 2012*). This type of understanding corresponds with the four (4) dimensions of interoception theory, based on which, the following stages of interoception can be designated: occurrence of afferent signals (1); encoding, integration and generation of representation (2); influence on cognitive and behavioural processes (3), a conscious expression of representation of inner bodily signals (4) (*Quadt, Critchley & Garfinkel, 2018*). Generally, people differ in terms of their ability to detect and properly report inner body signals. Thus, the question arises of how to assess interoceptive awareness? The concept of interoceptive awareness understood as an ability to detect, interpret and integrate inner bodily sensations is the foundation of the self-reported measurement tool, the Multidimensional Interoceptive Awareness Questionnaire developed by Mehling and coworkers (*Mehling et al., 2018*, *2012*). The questionnaire was applied in the current study as it is enables individuals to make a self-description in terms of interoceptive awareness linked with emotions, consciousness and behaviour (*Mehling et al., 2013*). The tool is built of eight scales addressing such components of interoceptive awareness as: Noticing; Not-Distracting; Not-Worrying; Attention Regulation; Emotional Awareness; Self-Regulation; Body Listening; and Trusting. The use of self-reported measurements also has some limitations. Recent research indicates that people express a limited ability to accurately reason about their inner bodily states, such as current cardiac activity (*Iodice et al., 2019*).

A more detailed description of the questionnaire is provided in the Materials & Methods section.

After the definitions of interoception have been outlined, we would like to discuss the relationship between interoceptive awareness and emotions. Firstly, the role of interoception understood in the narrow sense as a perception of the physiological circumstances of the body, will be discussed. Interoception is the basis of subjective emotional experience (*Calì et al., 2015*). Many theories of emotion accentuate the role of perception and interpretation of interoceptive signals (*Damasio, 1994*; *Schachter & Singer, 1962*). The exploration of the relationship between emotions and interoception became a topic of frequent interest to researchers (*Calì et al., 2015*). For instance, Damasio and colleagues emphasised the role of the representation of bodily experience in the brain, which is associated with specific emotions, subsequently affecting a decision making process (*Damasio, 1998*, *1994*; *Damasio et al., 2000*). It was empirically demonstrated that individuals who are more sensitive interoceptively experience emotions more intensively (*Montoya, Schandry & Müller, 1993*; *Pollatos, Kirsch & Schandry, 2005*; *Pollatos, Gramann & Schandry, 2007*; *Wiens, Mezzacappa & Katkin, 2000*). For example, research has shown that individuals highly aware of cardiac activity demonstrate a significantly stronger cardiac reaction to pleasant and unpleasant stimuli compared to less aware individuals (*Pollatos & Schandry, 2008*). Interestingly, interoceptive awareness seems to be associated with perception and appraisal of the arousal dimension (see *e.g.*, *Barrett et al., 2004*; *Dunn et al., 2010*). Moreover, higher interoceptive awareness was associated with greater recognition of affectively stimulating pictures, which might prove that individuals characterised by higher interoceptive awareness process emotionally arousing materials in a more conscious way (*Pollatos & Schandry, 2008*). *Calì et al. (2015)* suggested that interoception might influence emotional processing through a trait called emotional susceptibility, understood as a vulnerability and propensity to experience discomfort in response to emotionally charged stimuli. The proposed mechanism explaining the relationship between interoception and emotional experience seems to be supported by growing evidence of the role of the insula in interoceptive and emotional processes (*e.g.*, *Ernst et al., 2013*; *Zaki, Davis & Ochsner, 2012*). Taking into account the data quoted, it is not surprising that, in psychopathology body awareness was initially linked with excessive attention towards bodily sensations (or even symptoms) resulting in somatisation, rumination and catastrophic beliefs (*Cioffi, 1991*).

When considering interoceptive awareness in the wide sense, it is worth mentioning the study by *Calì et al. (2015)* once again. Research revealed that of the dimensions of the MAIA questionnaire (Attention Regulation, Trusting, Not-Worrying and Emotional Awareness) predicting emotional susceptibility (*Calì et al., 2015*), only Emotional Awareness is positively correlated with emotional susceptibility, whereas the rest of the scales are negatively correlated. A mindful, accepting experience of bodily sensations is a fundamental aspect of embodiment (*Carruthers, 2008*; *Fogel, 2009*). Many studies have shown that there is a link between mindfulness and interoception (*Farb et al., 2015*; *Hölzel*

*et al., 2011*; *Mehling et al., 2012*; *Tang, Hölzel & Posner, 2015*). In essence, both concepts can be theoretically and functionally treated as intertwined (*Farb et al., 2015*; *Hölzel et al., 2011*): mindfulness offers practices directly engaging interoceptive awareness and is based on the interoception mechanism as an intervention (*Bornemann & Singer, 2017*; *Farb et al., 2015*; *Hölzel et al., 2011*). Research has revealed that both IA and dispositional mindfulness predict psychological well-being (*Hanley, Mehling & Garland, 2017*).

Summing up the empirical data cited, the conclusion is that both beliefs about health and body and interoceptive awareness are associated with emotional experience. The relationship seems to be rather complex and further research in this area seems necessary.

## Aim and hypothesis

The current study seeks to address the understanding of the nature of the relationship between the types of variables: a self-reported emotional state classified on the valence and origin scales, dimensions of interoceptive awareness (*Mehling et al., 2012*, *2018*) and negative beliefs about health and body.

At first, the investigation was narrowed down to dimensions of interoceptive awareness and all types of emotions: negative automatic and reflective, positive automatic and reflective.

Therefore, we predicted that there is an association between two sets of variables: (I) all types of emotions and (II) the level of interoceptive awareness (H1). To examine multivariate networks of association between interoceptive awareness and different types of emotions, a canonical correlation analysis was planned.

Secondly, the aim of the study was to verify the relationship between negative beliefs and the emotions experienced. We expected that negative beliefs are positively associated with negative automatic and negative reflective emotions (Neg. Aut. Emo.; Neg. Refl. Emo.) (H2a). By contrast, negative beliefs are predicted to be negatively related to both automatic and reflective positive emotions (H2b).

We predicted also that there is generally a negative association between the MAIA scales and negative beliefs (H3).

Finally, the aim of the study was to demonstrate multiple distinct associations between all variables investigated. Research indicates that interoception might predict the intensity of emotional experience (see *e.g.*, *Dunn et al., 2010*; *Pollatos, Kirsch & Schandry, 2005*; *Terasawa et al., 2014*). Analogically, taking into account the ABC Ellis (*Ellis, 1991*) model used in cognitive or REBT therapy, the type and intensity of emotions experienced can to some extent be explained by referring to the subjective beliefs of the individual. Accordingly, based on the current state of knowledge, we assumed that the direction of the relationship runs from negative beliefs and MAIA scales to emotions. In other words, both MAIA scales and Neg. beliefs were expected to predict the intensity of Neg. Aut. Emo, Neg Refl. Emo, Pos. Aut. Emo and Pos. Refl. Emo (H4).
## MATERIALS AND METHODS

### Participants

The participants were volunteers recruited from different types of Facebook groups: groups associating local communities such as the inhabitants of big cities (Warsaw, Cracow, Trójmiasto and others), smaller cities and villages (like Mińsk Mazowiecki, Legnica, Ząbki and others) and regions (for example, *masovian voivodeship*, *lubelskie voivodeship*, *świętokrzyskie voivodeship*); groups of different university students (for example, the University of Warsaw, Warsaw Medical University, the Economic University in Cracow); and groups dedicated to different hobbies, like sport, health, self-development and psychology (but not addressed to professional psychologists). Group selection was designed to maximise sample diversification and avoid testing only the student population.

All Facebook groups are public and open to new members; however, some of them require filling in an additional survey to join the group. Members of the groups were asked to fill in an online survey consisting of several tools. The data was collected from 27 March to 15 April. During that time, about 50 posts were published with announcements encouraging individuals to take part in the study. This resulted in the receipt of more than 300 fully completed surveys. There was no remuneration for participation in the study.

The survey was completed by 327 Polish adults. The criteria applied to prepare the dataset were to exclude the responses wherein time of completion of the whole set of questionnaires differed by more than two standard deviations from the mean ($M = 7.92$ min; $SD = 3.76$; the time range for completing the study was from 2.68 min to 28.13 min). Based on that, 28 subjects' responses were removed. The final participant number was $N = 299$ (236 women, 62 men and 1 other), aged between 18 and 67 years old ($M = 30.41$; $SD = 10.96$).

No personal data was collected that would allowing us to identify participants. The demographic characteristics of the sample are included in Appendix 1.

### Design

The study was held in a correlational schema. There were three variables measured: (1) emotions experienced; (2) interoceptive awareness; and (3) negative beliefs about health and body.

### Materials

#### *Types of emotions*

Several tools were applied to measure the variables that were investigated. The first was a measurement of the emotions that had been experienced, consisting of two scales, one referring to negative emotions and the other referring to positive emotions (each scale containing 10 items). A mean was calculated from each scale. The measurements describing emotional state also used the origin dimension, and therefore, means from four scales were also calculated (each scale containing five items). The list was created based on the distinction between automatic and reflective emotions (*Imbir, 2016*), and therefore,

**Table 1 Emotion typology.** The table presents the typology of emotions based on valence and origin.

| Negative emotions | | Positive emotions | |
|---|---|---|---|
| **Automatic** | **Reflective** | **Automatic** | **Reflective** |
| Suffering | Shame | Alleviation | Persistence |
| Helplessness | Sadness | Reassurance | Compassion |
| Frustration | Dejection | Excitation | Hope |
| Breakdown | Depression | Relaxation | Courage |
| Terror | Envy | Tenderness | Loyalty |

further analysis was based on the division of emotions not only by valence, but also by origin.

Participants were asked to rate to what extent they felt the emotion in question at that moment. A seven-point scale was used, on which 1 means "to a small extent" and 7 means "to a significant extent".

Among negative emotions, the automatic ones are: Suffering (Polish: *Cierpienie*); Helplessness (*Bezradność*); Frustration (*Frustracja*); Breakdown (*Załamanie*); and Terror (*Przerażenie*). Negative reflective emotions include: Shame (*Wstyd*); Sadness (*Smutek*); Dejection (*Przygnębienie*); Depression (*Depresja*); and Envy (*Zawiść*).

The positive emotions were divided into automatic ones as well: Alleviation (*Ukojenie*); Reassurance (*Uspokojenie*); Excitation (*Ekscytacja*); Relaxation (*Odprężenie*); and Tenderness (*Czułość*). The reflective emotions are: Persistence (*Wytrwałość*); Compassion (*Współczucie*); Hope (*Nadzieja*); Courage (*Odwaga*); and Loyalty (*Lojalność*).

The choice of emotions had a theoretical and content-related character. Both the negative and positive emotions scales reached a good degree of reliability (negative emotions (Cronbach's alpha = 0.90); positive emotions (Cronbach's alpha = 0.77)). Similarly, origin-based scales also reached a good reliability (negative automatic emotions (Cronbach's alpha = 0.85); negative reflective emotions (Cronbach's alpha = 0.77); positive automatic emotions (Cronbach's alpha = 0.74); positive reflective emotions (Cronbach's alpha = 0.65)). The summary of emotion types is shown in Table 1. Means and standard deviations for emotion categories are shown in the summary Table 2.

### *Multidimensional assessment of interoceptive awareness*

In the current study, the MAIA questionnaire was used. There are eight scales within the tool which correspond to an eight-factor structure. The following scales were distinguished: Noticing; Not-Distracting; Not-Worrying; Attention Regulation; Emotional Awareness; Self-Regulation; Body Listening; and Trusting. Each scale is built from 3–7 items. The internal consistency of subscales ranges from 0.66 to 0.87, and inter-scale correlations range from 0.09 to 0.60 (*Mehling et al., 2012*). The current study used the Polish adaptation of the MAIA questionnaire (*Mehling et al., 2012*), developed by Anna Brytek-Matera and Anna Kozieł. The questionnaire scored very good reliability

**Table 2  Means, standard deviations and internal consistencies.**

|  | MEAN | SD | α |
|---|---|---|---|
| Negative automatic emotions | M = 3.53 | SD = 1.52 | 0.85 |
| Negative reflective emotions | M = 2.91 | SD = 1.25 | 0.77 |
| Positive automatic emotions | M = 3.01 | SD = 1.20 | 0.74 |
| Positive reflective emotions | M = 4.16 | SD = 1.12 | 0.65 |
| Noticing | M = 4.43 | SD = 1.03 | 0.69; *0.69* |
| Not Distracting | M = 3.34 | SD = 1.05 | 0.59; *0.66* |
| Not-Worrying | M = 3.12 | SD = 1.09 | 0.64; *0.67* |
| Attention Regulation | M = 3.65 | SD = 0.99 | 0.85; *0.87* |
| Emotional awareness | M = 4.54 | SD = 1.11 | 0.84; *0.82* |
| Self-Regulation | M = 3.21 | SD = 1.19 | 0.82; *0.83* |
| Body listening | M = 3.35 | SD = 1.28 | 0.81; *0.82* |
| Trusting | M = 3.89 | SD = 1.35 | 0.89; *0.79* |
| General beliefs about health and body | M = 2.58 | SD = 0.72 | 0.81 |

Note:
Means, standard deviations and internal consistencies for: emotions based on valence and origin emotion categories, MAIA scales, negative beliefs about health and body. For MAIA's scales, the results were compared with the reliability obtained in *Mehling et al. (2012)*, values are shown below in italics.

(Cronbach's alpha = 0.89). The scales' reliabilities (Cronbach's alpha) achieved values from 0.59 to 0.89. The results were compared with the reliability obtained in Mehling and colleagues' research (*Mehling et al., 2012*) and are shown in the summary Table 2 with means and standard deviations.

### Negative beliefs about health and the body

The last tool applied in the current study was the list of negative beliefs about health and the body in the epidemic context. The scale was created for the purpose of this research to consider ongoing circumstances. To verify how people's ways of thinking about their health and bodies determine their emotional state, a list of ten negative beliefs was formulated. The list, alongside a translation, is included in Table 3. The statements refer to perceived threats to life and health and body resistance–including the perceived efficiency of respiratory and immune systems.

To verify if a set of beliefs creates a one factor scale, a confirmatory factor analysis was run. The factorial structure of the scale was verified with the use of confirmatory factor analysis based on the maximum likelihood method. A unidimensional structure of the scores was verified. The fit of the unidimensional model was adequate after allowing for additional correlations between consecutive items no. 1 and no. 2, $r = 0.66$, $p < 0.001$, items no. 6 and no. 7, $r = 0.38$, $p < 0.001$, items no. 9 and no. 10, $r = 0.13$, $p < 0.05$, as well as between items no. 8 and no. 10, $r = 0.20$, $p < 0.001$, items no. 1 and no. 6, $r = 0.12$, $p < 0.003$, and items no. 4 and no. 8, $r = 0.15$, $p < 0.05$. The values of fit indices were equal to $CFI = 0.97$, $NFI = 0.94$, $RMSEA = 0.06$. Table 4 presents values of factor loadings acquired in the analysis.

The alpha reliability coefficient for the total scale was high and equal to 0.81.

**Table 3 Examples of negative beliefs about health and body used in the study.**

| Belief in English: | Polish translation: |
|---|---|
| 1. I believe that my health is in danger. | *Uważam, że moje zdrowie jest zagrożone.* |
| 2. I believe that my life is in danger. | *Uważam, że moje życie jest zagrożone.* |
| 3. My immune system is not durable enough, to deal with the disease. | *Mój układ odpornościowy nie jest wystarczająco silny, aby poradzić sobie z chorobą.* |
| 4. My body is weak. | *Moje ciało jest słabe.* |
| 5. I'm easily infected. | *Łatwo się zarażam.* |
| 6. To detect potential symptoms as quickly as possible, you need to carefully monitor your body's functioning. | *Aby jak najszybciej wykryć potencjalne objawy, trzeba uważnie przyglądać się funkcjonowaniu swojego organizmu.* |
| 7. Even a small change in the functioning of my body can be a symptom of a developing disease. | *Nawet niewielka zmiana w funkcjonowaniu mojego organizmu może być symptomem rozwijającej się choroby.* |
| 8. I can't do anything to protect my body from getting sick. | *Nie jestem w stanie nic zrobić, aby uchronić swój organizm przed zachorowaniem.* |
| 9. My respiratory system is not as efficient as for other people's respiratory systems. | *Mój układ oddechowy nie jest tak wydolny jak u innych osób.* |
| 10. If I fall ill, my body will definitely not be able to handle it. | *Jeżeli zachoruje, mój organizm na pewno sobie z tym nie poradzi.* |

**Table 4 Factor loadings acquired in confirmatory factor analysis.**

| Item no. | $f$ | $p$ |
|---|---|---|
| 1 | 0.48 | 0.001 |
| 2 | 0.49 | 0.001 |
| 3 | 0.74 | 0.001 |
| 4 | 0.76 | 0.001 |
| 5 | 0.67 | 0.001 |
| 6 | 0.28 | 0.001 |
| 7 | 0.34 | 0.001 |
| 8 | 0.30 | 0.001 |
| 9 | 0.58 | 0.001 |
| 10 | 0.47 | 0.001 |

Note:
$f$–factor loadings; $p$–statistical significance.

## Procedure

The data was collected from 27 March to 15 April. The recruitment procedure consisted of publishing announcements encouraging individuals to take part in the study. The message contained information about the expected time needed to complete the survey, the general purpose of the study, information about the anonymous nature of the study and appeals encouraging people to participate in the study. The only excluding criterion was age: the study was only addressed to adult participants. No data that would allow us to identify participants was collected during the study. The study was approved by Ethics Committee of the Faculty of Psychology in the University of Warsaw.

The study consisted of informed consent to participate in the study and several questionnaires, ordered as follows: measurement of emotions experienced; MAIA; beliefs

about health and body in relation to the ongoing epidemic; a demographic survey. The informed consent included detailed information about who obtains the data and for what purpose, information that the data is anonymously analysed for academic purposes only and that there are no data permitting identification of the participants. Additionally, the consent contained a request to fill out the questionnaires honestly and carefully, as there are no right or wrong answers. To confirm consent to participate in the study, participants were asked to click the "Go further" button. The consent was part of an online study; therefore it was given as an online approval, not in writing.

The study was conducted *via* the Qualtrics platform. Each tool was displayed on a separate screen. The first part was the assessment of emotional reaction. Participants were asked to evaluate on a seven-degree Likert scale (where 1 = not much, and 7 = very much) how much they currently experience each emotion. There were 20 emotions in total, all described above in the section on tools. Then participants filled out the MAIA questionnaire. Afterwards, participants were asked to assess how much they agree with the statements describing some beliefs about health and the body in the epidemic context. There were 10 statements for assessment. The last part was the demographic survey. The following data were collected *via* this survey: age, gender, place of residence, education and financial status (*i.e.*, if someone has a stable source of income and their perception of their financial status); state of health (*i.e.*, if someone suffers from a serious illness); and experiences related to the COVID-19 pandemic, like knowing someone who has had the disease and having the virus oneself. The demographic survey contained nine questions in total, most of which were closed.

## Statistical procedure

The statistical approach was aimed at verifying the relationship between three sets of variables: interoceptive awareness measured with the MAIA questionnaire, a negative belief about health and body operationalised with ten statements and four types of emotions based on the valence-origin division. For this purpose, a canonical correlation analysis (CCA) was applied to examine the multivariate structure of the association between the components of interoceptive awareness and different types of emotions experienced (automatic negative emotions, reflective negative emotions, positive automatic emotions, reflective automatic emotions). To illustrate the relationship between all variables investigated, a path analysis was also run. This type of analysis enables one to build a model including various dependencies between variables.

Additionally, some preliminary analyses were performed, such as a t-Student comparison based on the demographic factors, correlation analysis and regression analysis.

## RESULTS

In the first step, the relationship between the demographic data and all types of emotions was verified. The results are included in the Appendix.

As a part of the preliminary analysis, a correlation analysis was performed for the following pairs of variables: all types of emotions and the MAIA scale, all types of emotions

and negative beliefs about health and body, the MAIA scales. A summary of statistically significant results is attached in the Appendix.

Additionally, a multiple block regression was run. The explanatory models were run separately for all types of emotions. Demographic variables, MAIA scales and negative beliefs about health and body were entered in separated blocks. A comparison of the predictors obtained in the final models is included in the Appendix.

## Canonical correlations: the relationship between the MAIA scales and automatic and reflective emotions

To investigate the multivariate shared association between two sets of variables–the components of the MAIA questionnaire and variables based on the valence and origin types of emotions–a canonical correlation analysis was conducted. The full model was statistically significant using Wilk's $\lambda$ = .63, $F(32, 1,060.00)$ = 4.41, $p < 0.001.$, and explained 41% of the shared variance between the two sets of variables (see Fig. 1). Four functions were investigated using squared canonical correlations corresponding to respectively: (1) Rc2 = 0.27, (2) Rc2 = 0.11, (3) Rc2 = 0.03, (4) Rc2 = 0.01. However, only two of them were statistically significant. The first [27%; $F(31, 1,060)$ = 4.41, $p < 0.001$] and second [11%; $F(21, 827.53)$ = 2.16, $p = 0.002$] functions explained a substantial portion of shared variance. In the first function, two scales of the MAIA questionnaire, Self-Regulation and Trusting, were principally associated with three types of emotions: Neg. Aut. Emo., Neg. Refl. Emo., Pos. Aut. Emo. The sign of the correlation index for the Self-Regulation and Trusting scales was positive, the same as the correlation index in the case of the Pos. Aut. Emo., and contrary to Neg. Aut. Emo and Neg. Refl. Emo. In the second function, two scales of the MAIA questionnaire, Noticing and Not-Worrying, were primarily related with Pos. Refl. Emo. The correlation index for the Noticing scale and for Pos. Refl. Emo was minus, whereas the correlation index for the second scale, Not-Worrying, obtained a positive sign. All standardised canonical coefficients were reported in Fig. 1.

## Relationship between all types of emotions, negative beliefs about health and the body and the MAIA scales

The relationships between the study variables were analysed with the use of path analysis based on the maximum likelihood method. The preliminary model is depicted in Fig. 2.

The preliminary model differed significantly from the data that was analysed, $\chi^2(13)$ = 87.06, $p < 0.001$. The values of fit indices were $NFI$ = 0.91; $CFI$ = 0.92, $RMSEA$ = 0.14. In order to improve the fit of the model, statistically insignificant paths were removed, and statistically significant paths were added. The addition of paths was based on the values of modification indices with a threshold value of 4. The path that was statistically insignificant was the path between Attention Regulation and Emo. Pos. Refl., $B$ = 0.12, $p > 0.05$. The only directional relationship involving Attention Regulation was the path between Attention Regulation and Emo. Pos. Refl. Therefore, Attention Regulation was excluded from the model. In the next three consecutive steps correlations between Emo. Neg. Aut. and Emo. Pos. Aut., between Emo. Neg. Refl. and Emo. Pos. Aut.
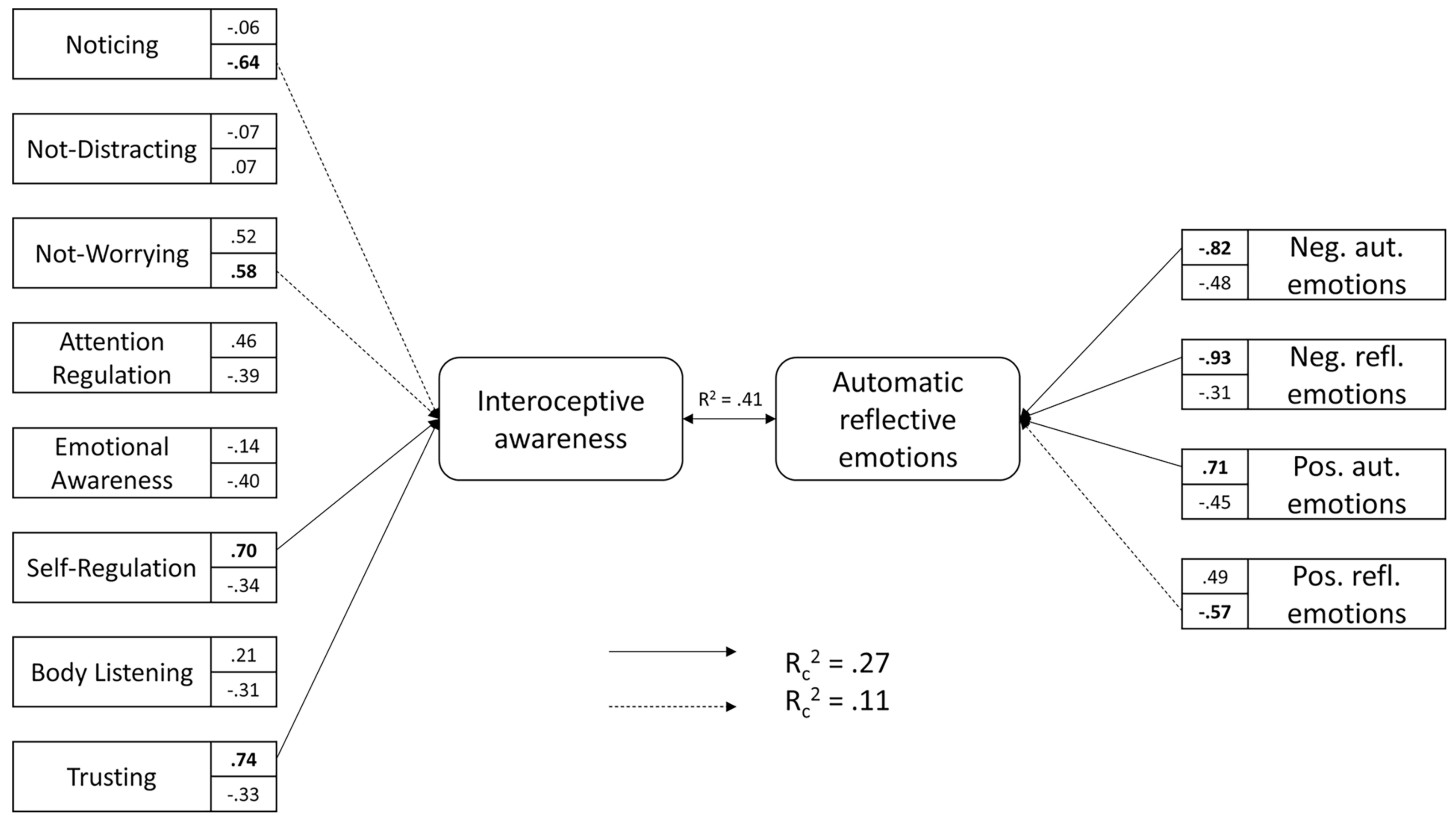

**Figure 1 Standardized canonical coefficients for the first and second functions.** Each box contains the standardized canonical coefficients for the variable in the first function on the top, and for the second function on the bottom. The first function is represented by a solid line, and the second function is represented by dashed lines. Shared variance among interoceptive awareness and automatic and reflective emotions across all functions is described with R2. Squared canonical correlations representing shared variance between MAIA's scales and automatic and reflective emotions are described with Rc2 for each function.

and a path between Trusting and Emo. Pos. Refl. were added. After these four modifications the final model depicted in Fig. 3 did not differ significantly from the data that was analysed, $\chi^2(6) = 9.70$, $p > 0.05$. The values of fit indices were $NFI = 0.99$; $CFI = 0.99$, $RMSEA = 0.05$.

Table 5 presents the values of the correlation coefficients. Table 6 presents the values of the regression coefficients.

Not-Worrying was positively correlated with Self-Regulation, Self-Regulation was positively correlated with Trusting and Trusting was positively correlated with Not-Worrying. Negative beliefs correlated negatively with both Trusting and Not-Worrying. The indicators of positive emotions showed a positive correlation with each other. The indicators of negative emotions also showed a positive correlation with each other. The indicators of positive emotions and the indicators of negative emotions correlated negatively with each other.

Self-Regulation, Trusting and Not-Worrying were negatively related to Emo. Neg. Aut. and Emo. Neg. Refl. Negative beliefs were related to Emo. Neg. Aut. and Emo. Neg. Refl. positively. Self-Regulation and Trusting were positively related to Emo. Pos. Aut. and Emo. Pos. Refl. Negative beliefs were related to Emo. Pos. Aut. negatively. The final
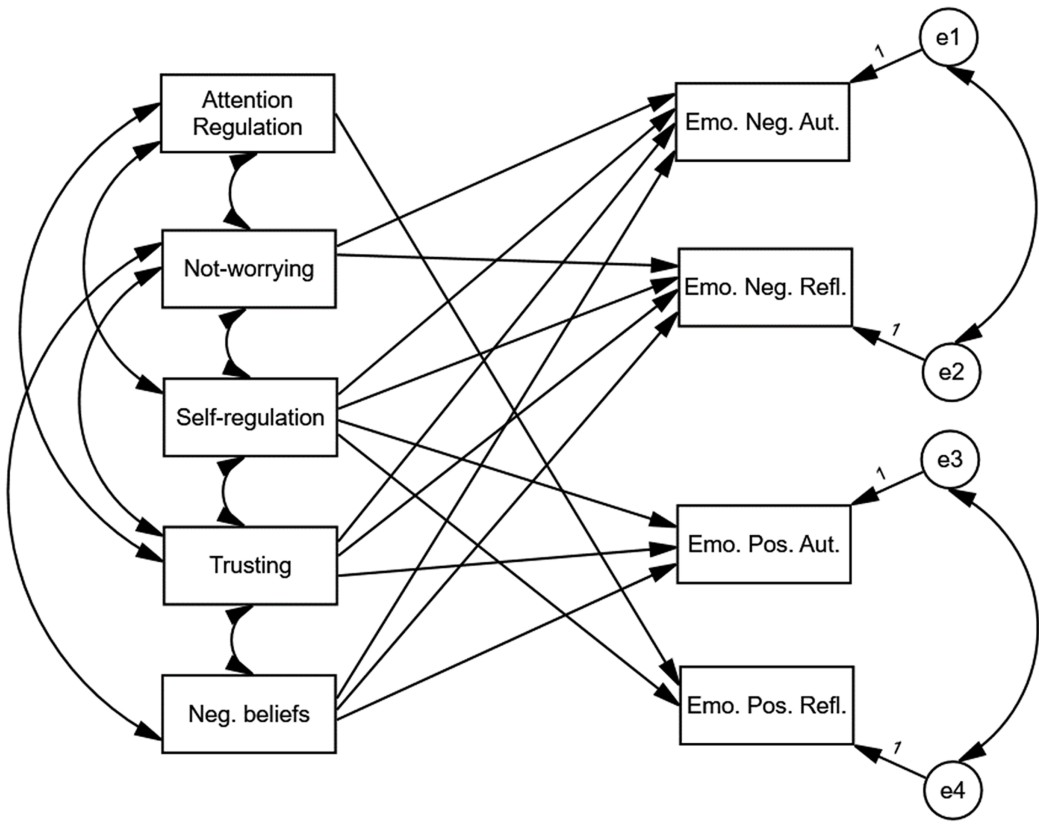

**Figure 2 The preliminary model of relationships between different types of emotions, MATA's scales and negative beliefs about health and body.** The preliminary model differed significantly from the analysed data, $c^2(13) = 87.06$, $p < 0.001$. The values of fit indices were equal to *NFI* = 0.91; *CFI* = 0.92, *RMSEA* = 0.14.

model explained 7.8% of Emo. Pos. Refl. Variance, 15.2% of Emo. Pos. Aut. variance, 25.8% of Emo. Neg. Refl. variance and 24.5% of Emo. Neg. Aut variance.

# DISCUSSION

## The relationship between different types of emotions and interoceptive awareness

The current study showed that there is a multivariate shared association between two sets of variables: MAIA components and different types of emotions: Neg. Aut. Emo., Neg. Refl. Emo, Pos. Aut. Emo., Pos. Refl. Emo. The model explained 41% of the shared variance between two sets of variables. Scales such as: Noticing, Not-Worrying, Self-Regulation, Trusting were primarily associated with emotions. The canonical correlation analysis delivered interesting data illustrating a multivariate association between automatic and reflective emotions and interoceptive awareness. The first function, explaining the majority of the common variance, indicated a strong relationship between the Self-Regulation and Trusting scales and emotions–with the increase of those dimensions in the function, the decrease of negative automatic and negative reflective emotions occurs, whereas positive automatic emotions escalate. An interesting pattern of relation between MAIA

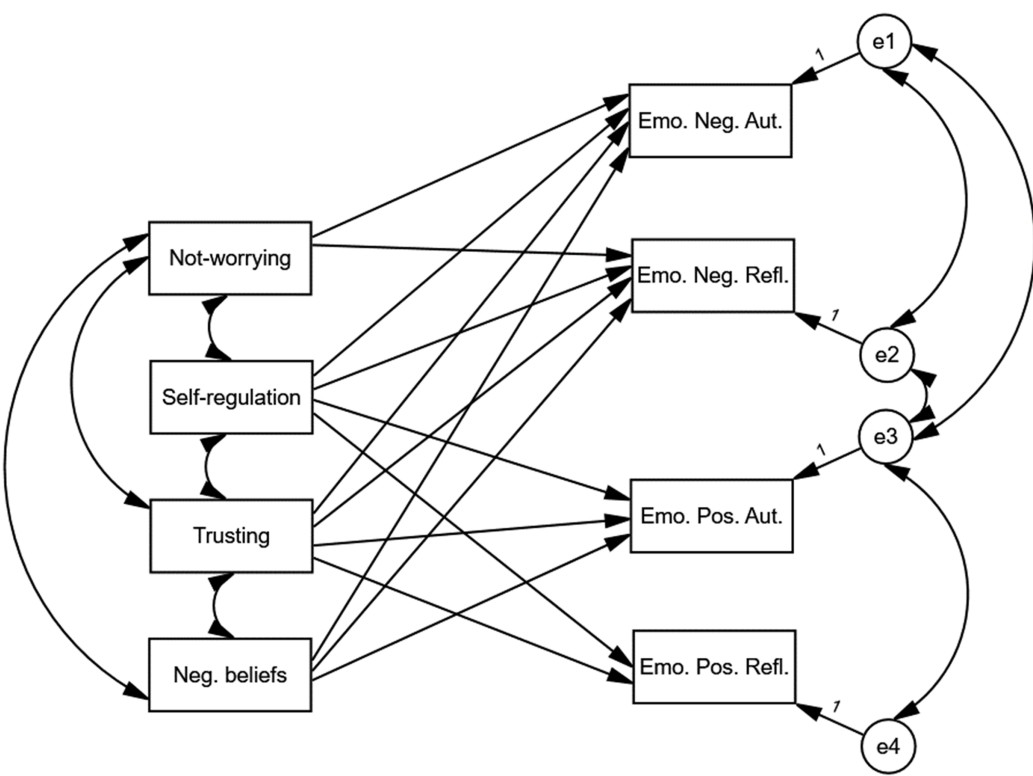

**Figure 3 Final model of relationships between different types of emotions, MAIA's scales and negative beliefs about health and body.** The final model depicted did not differ significantly from the analysed data, $\chi^2(6) = 9.70$, $p > 0.05$. The values of fit indices were equal to *NFI* = 0.99; *CFI* = 0.99, *RMSEA* = 0.05.

**Table 5 Values of correlation coefficients in the model.**

| Relationships | | | *r* | *p* |
|---|---|---|---|---|
| Not-worrying | <–> | Self-regulation | 0.15 | 0.008 |
| Self-regulation | <–> | Trusting | 0.48 | 0.001 |
| Trusting | <–> | Neg. beliefs | −0.11 | 0.027 |
| Not-worrying | <–> | Trusting | 0.15 | 0.011 |
| Not-worrying | <–> | Neg. beliefs | −0.32 | 0.001 |
| Emo. Neg. Aut. | <–> | Emo. Neg. Refl. | 0.79 | 0.001 |
| Emo. Pos. Aut. | <–> | Emo. Pos. Relf. | 0.37 | 0.001 |
| Emo. Neg. Aut. | <–> | Emo. Pos. Aut. | −0.41 | 0.001 |
| Emo. Neg. Refl. | <–> | Emo. Pos. Aut. | −0.29 | 0.001 |

**Note:**
  *r*–correlation coefficient; *p*–statistical significance.

and emotions was investigated through the second function in canonical correlation analysis–there is an association between Noticing, Not-Worrying and positive reflective emotions. The increase of Noticing leads to an increase of positive reflective emotions, whereas an increase of Not-Worrying was associated with the opposite observation. The Noticing scale reflects the subject's ability to detect inner body symptoms without

**Table 6 Values of regression coefficients in the model.**

| Relationships | | | B | p |
|---|---|---|---|---|
| Emo. Neg. Aut. | <— | Self-regulation | -0.13 | 0.031 |
| Emo. Neg. Aut. | <— | Trusting | -0.12 | 0.032 |
| Emo. Neg. Aut. | <— | Neg. beliefs | 0.34 | 0.001 |
| Emo. Neg. Aut. | <— | Not-worrying | -0.17 | 0.001 |
| Emo. Neg. Refl. | <— | Not-worrying | -0.16 | 0.002 |
| Emo. Neg. Refl. | <— | Self-regulation | -0.17 | 0.004 |
| Emo. Neg. Refl. | <— | Trusting | -0.16 | 0.005 |
| Emo. Neg. Refl. | <— | Neg. beliefs | 0.31 | 0.001 |
| Emo. Pos. Aut. | <— | Self-regulation | 0.19 | 0.002 |
| Emo. Pos. Aut. | <— | Trusting | 0.22 | 0.001 |
| Emo. Pos. Aut. | <— | Neg. beliefs | -0.13 | 0.008 |
| Emo. Pos. Refl. | <— | Trusting | 0.17 | 0.008 |
| Emo. Pos. Refl. | <— | Self-regulation | 0.16 | 0.014 |

Note:
 $B$–regression coefficient; $p$–statistical significance.

psychological appraisal and integration; it refers to the elementary aspect of interoception (*Quadt, Critchley & Garfinkel, 2018*). The Not-Worrying scale contains statements expressing that the individual does not react with worry or upset on pain or discomfort. Only one statement refers directly to the ability to notice the sensations without concerns. Therefore, juxtaposing both scales, one can get the impression that some of them may express slightly opposite tendencies. However, the second function explains the minor part of the variance, therefore this result should rather be treated with caution.

## The structure of associations between all types of emotions, negative beliefs about health and the body and MAIA scales

The model illustrating the multiple distinct associations between negative beliefs, MAIA components and four types of emotions: Neg. Aut. Emo., Neg Refl. Emo., Pos. Aut. Emo., Pos. Refl. Emo. revealed a good adjustment to data and brought some new insight.

### *Non-directional relationships between the variables that have been tested*

There is a relationship between Not-Worrying and Trusting which seems to be consistent with the theory, as both scales refer to an accepting, fearless, reliant attitude towards inner body sensations. On the other hand, both scales are not explicitly intertwined in the factorial structure of the MAIA questionnaire (*Hanley, Mehling & Garland, 2017*; *Mehling et al., 2018*). Moreover, the relationship is weak. Interestingly, there is also a moderately strong relationship between the negative beliefs and Not-Worrying scales. Considering the content of both factors, it can be inferred that both components are theoretically opposite, as the Not-Worrying scale enables evaluation of attentional and emotional response to perceived bodily signals (*Hanley, Mehling & Garland, 2017*), whereas the negative beliefs scale reflects a negative appraisal on the body. Another non-directional relationship was observed between all types of emotions. We recognised a

strong relationship between negative automatic and negative reflective emotions, a moderate relationship between positive automatic and positive reflective emotions, and a moderate negative relationship between negative automatic emotions and positive automatic emotions. Finally, there was a low negative relationship between negative reflective emotions and positive automatic emotions. The results obtained can be interpreted in the context of the dimensions of emotions. The division by origin seems to be more efficient in terms of positive emotions than with to negative. The split based on valence is more compelling than the distinction based on origin.

### The role of negative beliefs about health and body in explaining of the intensity of emotions

Firstly, the model demonstrated a meaningful role of negative beliefs. There is an association between negative beliefs about health and the body and all types of declared emotions apart from Pos. Refl. Emo. Negative beliefs significantly predict the intensity of reported emotions–they are the strongest predictor for Neg. Aut. Emo. and Neg. Refl. Emo. Beliefs about being in danger and evaluation of one's body as weak seem to play a crucial role in the reported emotional state, especially in the framework of the COVID-19 pandemic. This result can be interpreted in the context of a cognitive-behavioural approach. For instance, in hypochondriasis, biases in thinking, such as negative automatic thoughts–instead of realistic, adaptive beliefs–influence affective reaction (see *e.g., Calvo & Dolores Castillo, 2001; Marcus et al., 2007; Williams, 2004*). Applying this view to current findings suggests that even an individual who does not suffer from an anxiety disorder, might experience increasing fear for their health in such an unpredictable situation as a pandemic, partly due to undermining, fearful beliefs.

### The role of MAIA components in explaining intensity of emotions

The path analysis revealed that scales such as Self-Regulation and Trusting predict all types of emotions. The Not-Worrying scale predicts only negative emotions.

The findings described indicate a crucial role of scales such as Self-Regulation, Trusting and Not-Worrying in the processes regulating the emotion, understood as minimising negative affect, which indirectly confirms the regulatory role of interoceptive awareness. The result remains consistent with the current state of knowledge, several studies have demonstrated that higher interoceptive awareness promotes affect downregulation through reappraisal, which is reflected in neural activity (*Füstös et al., 2012*). Moreover, individuals who are more aware of inner bodily sensations demonstrated more antecedent-focused and response-focused strategies for regulating emotion, such as supportive reappraisal or maladaptive suppression (*Kever et al., 2015*). Similar results were obtained in the game-playing context, where it was observed that greater interoceptive awareness was related to regulation strategies for emotions relating to health such as attempting a resolution of a negative state or seeking social support (*Lobel, Granic & Engels, 2014*).

The observed mechanism might be understood in the context of interoceptive awareness's relation with psychological well-being (see *Hanley, Mehling & Garland, 2017*).

Interoceptive awareness, together with dispositional mindfulness, predicts psychological well-being. It seems that enhancing the accepting body-mind relationship might deliver a more positive affect, especially in the case of automatic homeostatic emotions. However, more empirical data is required to verify this relation.

### Summary of the model structure

For Neg. Aut. Emo and Neg. Refl. Emo we observed the same structure of predictors. The structure of predictors for negative and positive emotions shows moderate differences. Therefore, the following conclusions can be drawn: (1) the valence dimension (positive *vs.* negative) differentiates the structure of explanatory models; in other words, different factors predict the intensity of positive emotions and of negative emotions; (2) the origin dimension differentiates the structure of predictors slightly and refers to the presence of negative beliefs.

The model explained the variance of negative emotions, both reflective (25.85%) and automatic (24.5%), to a greater extent than positive (15.2% of Emo. Pos. Aut. variance and 7.8% of Emo. Pos. Refl. variance). The above results indicate that the influence of factors related to experiencing and assessing one's own body occurs within the reduction of negative affect rather than the enhancement of positive affect.

## Limitations

This study has its limitations. The data were collected from users of Facebook groups. In the sample, most participants were inhabitants of large cities and persons with higher education and which, therefore, constrained the representativeness of the sample. For that reason, the results may be a specific to the population, presenting relatively high levels of interoceptive awareness. Further studies on the general population are needed. The sample size can also be considered as a study limitation, especially when data is collected *via* the Internet. Due to specific time-related restrictions regarding movement, the only way to collect data was to conduct an online survey. This method excludes the use of objective measures of interoception, such as heartbeat tracking tasks, which in turn could deliver interesting information.

To understand more profoundly the character of the relationship between one's bodily experience, negative beliefs about health and the body and emotional responses to threatening events, it seems useful to apply additional self-reported measurements, such as dispositional mindfulness, emotion regulation strategies and coping styles. In the study, only two dimensions of emotions were operationalised. The dimension of arousal was not included in the design due to the aim of the study (the purpose was to verify the relationship between the body related variable and the emotions based on variation in the valence and arousal dimensions). This direction of research seems to be interesting, as there is some evidence that highly interoceptively aware individuals perceive stimuli as more arousing (*Barrett et al., 2004*; *Dunn et al., 2010*). Moreover, there are varied approaches to emotions (*e.g.*, *Moors et al., 2019*); possibly applying another theoretical framework could result in a new direction of research.

## CONCLUSIONS

The study has revealed that both interoceptive awareness and negative beliefs about health and the body significantly explain the variance in emotional responses reported during the COVID-19 pandemic. Scales such as Self-Regulation and Trusting are predictors for the types of emotions that were tested. This result highlights the regulative role of interoceptive awareness. The trustful attitude towards the body and its capacity for regulating distress by directing attention to body sensations enables an improvement of individual well-being through diminishing the experiencing of negative emotions and enhancing the experiencing of positive ones. At the same time, Not-Worrying is only a valid predictor for negative emotions. However, in the case of negative emotions, it was negative beliefs about health and the body that turned out to be the strongest predictor. The emotion typology applied—based on the dimensions of valence and origin—brings about a wider understanding of the nature of emotional responses in general.

### Funding

This work was supported by the Faculty of Psychology, University of Warsaw, from the funds awarded by the Ministry of Science and Higher Education in the form of a subsidy for the maintenance and development of research potential in 2021. The funders had no role in study design, data collection and analysis, decision to publish, or preparation of the manuscript. The publication fee was financed from the funds from "Excellence Initiative—Research University (2020-2026)".

### Grant Disclosures

The following grant information was disclosed by the authors:
Ministry of Science and Higher Education in the form of a subsidy for the maintenance and development of research potential in 2021.
Excellence Initiative – Research University (2020-2026).

### Competing Interests

The authors declare that they have no competing interests.

### Author Contributions

- Aleksandra Modzelewska conceived and designed the experiments, performed the experiments, analyzed the data, prepared figures and/or tables, and approved the final draft.
- Kamil K Imbir analyzed the data, authored or reviewed drafts of the paper, and approved the final draft.

### Human Ethics

The following information was supplied relating to ethical approvals (*i.e.*, approving body and any reference numbers):

The study was approved by Research Ethics Committee on Faculty of Psychology at University of Warsaw.

## Data Availability

The raw data is available in the Supplemental Files.

## Supplemental Information

Supplemental information for this article can be found online at http://dx.doi.org/10.7717/peerj.12542#supplemental-information.

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
