# Peer review of "Interoceptive awareness and beliefs about health and the body as predictors of the intensity of emotions experienced at the beginning of the pandemic"

_PeerJ, doi:10.7717/peerj.12542_

## Round 0.1 · original submission · Major Revisions

With the comments in hand, I must condition the acceptance of the current manuscript on major revisions, as requested by the two Reviewers.

In fact, the two reviewers point at some important flaws that need to be addressed. I will just cite a few, but I recommend responding to each of their comments.

For starters, I agree with Reviewer #1 and #2 on the fact that the introduction seems to be lengthy, and it focuses on topics that are not directly addressed by the paper. On the other hand, I was left with the impression that some of the focal core hypotheses have been poorly introduced (line 182-185, see also Reviewer #2 comments), and a lot of definitions need to be provided (see Reviewer #1 comments).

Reviewer #2 offers some useful suggestions on relevant literature that could be referred to in order to improve the introduction and to widen the perspective on the different theoretical accounts on emotions. This literature is also useful for improving the discussion.

The analytical approach does not seem convincing, and I concur with Reviewer #1 recommendation of conducting an SEM approach to take measurement models into account and to test different hypotheses about different paths.

Reviewer # 1 notes that the MAIA-COVID connection is a bit far fetched, and I agree with this assessment.

Here the authors can find some additional comments from myself:

- A confirmatory factor analysis, rather than an exploratory factor analysis (by the way, a PCA is not even a proper factor analysis, to be picky)

- It is not clear why the authors used a non-parametric test for comparing means, and a parametric one when looking at correlations. Given the sample size, and therefore the applicability of the central limit theorem, I think that the use of a parametric t-test for comparing means would be just as appropriate.

- I know it sounds pedantic, but it is not correct to use the word "prove" when describing empirical research and, even worse, when referring to correlative studies.

- Check for typos (e.g. "proceed" instead of "processes" in line 238, or stable instead of unstable in line 458).

Reviewer 1 ·

Basic reporting

In my opinion, the introduction is way too long. I believe it could benefit from a revision to increase the focus on the important concepts only. Nevertheless, I acknowledge this is also subject to editorial decision.

To avoid misinterpretations with other definitions of interoceptive awareness, it should be specified in the abstract that the concept of interoceptive awareness adopted refers to that proposed by Mehling and colleagues in the MAIA.

Lines 128- 129: The authors wrote "However, valence is more intuitive when classifying emotional experiences, and thus in the current research, we place a strong emphasis on the valence dimension". Buy it's not cleat to me why valence. Can't we assume that the vast majority will perceive this as negative, thus with little variation along this dimension, but the level of intensity or arousal will vary in greater degree? Please clarify or revise


line 273: define emotional susceptibility

The difference between hypothesis 1 and hypothesis 2 could be clearer (line 273).

Acknowledging I am not an expert on canonical correlations, is there statistics to be reported on the associations between specific MAIA dimensions and emotions (lines 543-547)?

For clarity, examples of the beliefs questions, specifying the precise phrasing, should be given in the main text

As a suggestion, the authors could consider a way of organizing and synthesizing all the measures, analyses and main findings, for example in a summary table.

A few sentences and expressions are grammatically awkward. The manuscript could benefit from proof-reading

There is no mention to ethics approval nor informed consent

Experimental design

The analyses plan consisted in several approaches to examine the relationship between 2 variables or 2 sets of variables (i.e. canonical correlations). However, because the data set comprises, at least, 3 sets of variables (i.e. emotions, beliefs, intereoceptive beliefs) wouldn't it make sense to try to investigate how these three sets are related to each other, for example through targeted or theoretically driven mediations/moderation analyses or Structural Equation Modelling. I feel the different analyses report "a lot of the same" and it would be interesting to go beyond this level of analyses. I would like to know the authors' thoughts on this.

I am not fully convinced the emotions questions, and the way in which they were framed, are sufficiently related to the Covid-19 pandemic. If that would be the case it would have important implications for the interpretation of results. However, I might be missing some detail and the authors might be able to convince me otherwise. Please clarify.

Validity of the findings

Further description of each statistical analysis and justification for its use is needed in the methods section. Three sets of analyses are carried out but little explanation is given for why these are necessary and how they differ between them. In fact, it was very difficult for me to follow the results section.As a reader, as I was going through the long "list" of results (and many tables) I felt there were really too many analyses that are conceptually similar. This meant that, for me, that it was difficult to follow through all the analyses and extract the (most) meaningful information. I advise, not only to provide greater justification for the analyses plan but also to consider removing, or at least moving it to supplementary materials, some of the analyses (e.g. simple correlations; demographics).

The manuscript is framed as directly related to Covid-19 but I am not fully convinced the methods and discussion of results truly reflect this. In addition to my comment above on the relevance of the emotions question, it's not clear to me that enough has been done in terms of analyses/discussion to make bold claims about reactions to Covid-19. For example, how does MAIA dimensions are related to feelings related to Covid-19. Also, not much attention in discussion and analyses (e.g. canonical correlations) was given to the beliefs variables and I'd think these would be the ones mostly related to Covid-19.

Reviewer 2 ·

Basic reporting

The study reported in manuscript identifies a relation among some dimensions concerning the Interoceptive awareness (Self-Regulation, Trusting and Not-Worrying dimensions) and the emotional reaction to the COVID-19 pandemic. Overall, the manuscript is well written and offers an interesting contribution to the understanding of the psychological/emotional experience of the pandemic.
I have some points here below:
I suggest the authors to reduce the introduction session, especially the paragraph from line 111 to 158, too many words are spent to talk about emotion and the literature mentioned is a bit old.
Lines 117: I would not say that emotions are reactions, emotions are goal-directed action tendencies. I invite the author to read and mention:

Moors, A., Boddez, Y., & De Houwer, J. (2017). The power of goal-directed processes in the causation of emotional and other actions. Emotion Review, 9(4), 310-318.

Moors, A., & Fischer, M. (2019). Demystifying the role of emotion in behaviour: toward a goal-directed account. Cognition and Emotion, 33(1), 94-100.

The valence and arousal are not the only dimensions in defining emotions.
In the goal-directed view, there are also: the experienced control of the aversive stimuli, the value attributed to the goal pursued and the expectancy about the reached outcome.
See Moors, A., Fini, C., Everaert, T., Bardi, L., Bossuyt, E., Kuppens, P., & Brass, M. (2019). The role of stimulus-driven versus goal-directed processes in fight and flight tendencies measured with motor evoked potentials induced by Transcranial Magnetic Stimulation. PloS one, 14(5), e0217266.
Moreover, I do not totally agree with the all-of-none view of automaticity, invoked to separate fast and dirty automatic emotions from more complex cognitive reflective emotions.
I invite the authors to read:
Moors, A., & De Houwer, J. (2006). Automaticity: a theoretical and conceptual analysis. Psychological bulletin, 132(2), 297.
In this theoretical paper, the authors provide a deep insight of the concept of automaticity and suggest to think about the automaticity as composed by multiple features which need to be separately investigated and to be considered along a continuum in all the cognitive processes, including the emotions. Basically, they invite all the researchers to overrule the all-or-none view.
Concerning the paragraph: “Factors related to psychological attitude towards the body affecting epidemic responses”, I invite the authors to synthetize the paragraph and to talk about interoception, by adding recent literature.
See:
Iodice, P., Porciello, G., Bufalari, I., Barca, L., & Pezzulo, G. (2019). An interoceptive illusion of effort induced by false heart-rate feedback. Proceedings of the National Academy of Sciences, 116(28), 13897-13902.
Quadt, L., Critchley, H. D., & Garfinkel, S. N. (2018). The neurobiology of interoception in health and disease. Annals of the New York Academy of Sciences, 1428(1), 112-128.
Pezzulo, G., Maisto, D., Barca, L., & Van den Bergh, O. (2019). Symptom perception from a predictive processing perspective. Clinical Psychology in Europe, 1(4), 1-14.

Experimental design

The Hypothesis of the study are not clear to me and I cannot follow the rationale which has led to declare them.
By taking into consideration the mentioned literature about the relation between interoceptive awareness and emotion susceptibility, I would aspect a more sophisticated hypothesis, maybe there are only some components of the interoceptive awareness associated with the diminished experience of negative emotion (H1).
Procedure:
The authors set as sample criteria just the age, I wonder whether it would be important asking about the use of medications which can alter the body sensations.

Validity of the findings

Results
Line 445: “Negative emotions are negatively related to perceived material status”, please reframe more clearly
Discussion:
Lines 660-662: “Interestingly, Noticing and Emotional awareness were positively related with negative emotions; however, the correlation was low.”
I invite the authors to spend some words to discuss this data which is consistent with part of the literature mentioned in the intro.

Additional comments

Check the typos

---

## Round 0.2 · Major Revisions

Dear authors,

As you can see, both reviewers agree that the manuscript has improved, but they both ask for further revisions.

In particular, both reviewers ask for some proofreading to check the soundness of some sentences.

Furthermore, Reviewer one raises their concerns about the redundancy of some of the reported analyses and asks for clarification about the emotions questionnaire.

Reviewer 1 ·

Basic reporting

no comment

Experimental design

no comment

Validity of the findings

no comment

Additional comments

I thank the authors for their effort to address my comments and appreciate the extensive revisions made. I believe the manuscript has improved considerably, especially in terms of readability. However, I still have some concerns I would like to see clarified or addressed.

I might be missing something but I still believe there is substantial conceptual repetition in the analyses performed. Path analyses does indeed seem to be a good addition to the manuscript and brought some additional depth to the study (please note that I am not an expert neither in canonical correlations nor path analyses). However, I am not sure what is the added value of reporting the simple correlations and the regressions models on top of the canonical correlations and path analyses. It is possible that I might be missing something and the authors might be able to convince me otherwise, but I’d think these are peripherical and should, at most be moved into the appendix. I am afraid I personally find the results section very difficult to digest with all the different (but conceptually similar) results. Unfortunately, this also transpires into the discussion section.
Also, if correlation analyses are to be reported these should be corrected for multiple comparisons.

Minor:
I would advice the authors to have the manuscript (including abstract) proof-read, ideally by someone in the field, as there are several awkward sentences and expressions. For example: “The COVID-19 pandemic is basically a strong stressor.” Also, I appreciate the authors effort to reduce and simplify the theoretical background in the introduction. However, in certain places the flow of information is somewhat fragmented and lacks fluidity (e.g. paragraph on stress and physiological reactions). I’d advise the authors to revise.

Hypothesis 2 seems to be missing

The description of the emotions questionnaire could be clearer. What precise question was asked? That is, were the question on felt emotions placed in the context of the pandemic or simply referred to “here and now”? These might lead to considerable different interpretations. The table added does clarify considerably some questions I had (mostly regarding how to measure the origin dimension) but a few details could still be clearer.

Please carefully review the results section as there might be several typos. For example, in line 483, should it read “negative emotions”?; in line 505, should it read “three of which”?.

Reviewer 2 ·

Basic reporting

Overall, I find the manuscript improved; as required, the authors removed some less relevant pieces of introductions and added other important references from the recent literature.
I kindly encourage the authors to revise/improve the writing in order to endorse the publication of the manuscript.

Experimental design

no comment

Validity of the findings

no comment

---

## Round 0.3 · Minor Revisions

One of the Reviewers has some additional concerns about the clarity of a paragraph.

Reviewer 2 ·

Basic reporting

Before endorsing the publication of the manuscript. I kindly invite the authors to re-phrase this paragraph:
"Attempts to understand the relationship between experiencing your body and emotions should also distinguish between the perception of sensations and their declared feeling Pezzulo
and colleagues (2019) indicate that physiological dysfunctions are associated with self-reported
symptoms, when an individual suffers from a disease. Authors explained the fluctuation in the
relationship between psychological dysfunctions affected by the disease and reported symptoms,
using the predictive processing perspective. The approach is based on exploration of the
construction of the adaptive model by the brain, thus referring to predictive coding revealed in
perception and an active interference action. This conceptualisation addresses the issue of pre existing (prior) information and its interference in the construction of newly occurring
representation, and therefore it offers an interesting perspective in understanding the role of
cognitive beliefs in experiencing a stressful situation."
It should be written more clearly, at the moment it appears quite obscure.

Experimental design

no comment

Validity of the findings

no comment

Additional comments

no comment

---

## Round 0.4 · accepted · Accept

I am glad to announce that I am satisfied with the Authors' response, and I now consider the paper suitable for publication.